# The Role of Physical Function in the Association between Physical Activity and Gait Speed in Older Adults: A Mediation Analysis

**DOI:** 10.3390/ijerph191912581

**Published:** 2022-10-01

**Authors:** Marcelo de Maio Nascimento, Élvio Rúbio Gouveia, Adilson Marques, Bruna R. Gouveia, Priscila Marconcin, Cíntia França, Andreas Ihle

**Affiliations:** 1Department of Physical Education, Federal University of Vale do São Francisco, Petrolina 56304-917, Brazil; 2Department of Physical Education and Sport, University of Madeira, 9020-105 Funchal, Portugal; 3Laboratory for Robotics and Engineering System (LARSYS), Interactive Technologies Institute, 9020-105 Funchal, Portugal; 4Center for the Interdisciplinary Study of Gerontology and Vulnerability, University of Geneva, 1205 Geneva, Switzerland; 5Interdisciplinary Centre for the Study of Human Performance (CIPER), Faculty of Human Kinetics, University of Lisbon, 1495-751 Lisbon, Portugal; 6Instituto de Saúde Ambiental (ISAMB), Faculty of Medicine, University of Lisbon, 1649-020 Lisbon, Portugal; 7Regional Directorate of Health, Secretary of Health of the Autonomous Region of Madeira, 9004-515 Funchal, Portugal; 8Saint Joseph of Cluny Higher School of Nursing, 9050-535 Funchal, Portugal; 9KinesioLab, Research Unit in Human Movement Analysis, Piaget Institute, 2805-059 Almada, Portugal; 10Department of Psychology, University of Geneva, 1205 Geneva, Switzerland; 11Swiss National Centre of Competence in Research LIVES–Overcoming Vulnerability: Life Course Perspectives, 1015 Lausanne, Switzerland

**Keywords:** aging, sedentary behavior, physical activity, physical function, mobility, vulnerability, older adults

## Abstract

**Simple Summary:**

Aging is associated with vulnerability in terms of a natural decline of most physiological systems, and, consequently, physical function (PF) performance (e.g., cardiorespiratory performance, muscle strength, flexibility, speed, balance) decreases. Adequate physical activity (PA) levels are essential to maintaining or increasing PF performance, directly influencing gait speed (GS). Having a fast GS increases the older adult’s capacity to perform daily tasks safely and remain autonomous at an advanced age. Our study aimed to explore the mediating role of PF in the relationship between PA and GS in a large sample of older adults from the north of Brazil. Regarding the PA-total level, the analysis showed that a fast GS was partially mediated by approximately 19% by a better PF performance. PF partially mediated the association between PA-sport and GS in approximately 9%, and PF partially mediated the association between PA-leisure and GS in 46%. We observed a significant and negative association between PA-housework and GS. Thus, PF partially mediated the association in about 9% of cases. Consequently, our study suggests that among older adults, PF plays a crucial role in mediating the association between PA and GS levels in the vulnerable aging population.

**Abstract:**

Adequate levels of physical function (PF) are essential for vulnerable older adults to perform their daily tasks safely and remain autonomous. Our objective was to explore the mediating role of PF in the relationship between physical activity (PA) and gait speed (GS) in a large sample of older adults from the north of Brazil. This is a cross-sectional study that analyzed 697 older adults (mean age 70.35 ± 6.86 years) who participated in the project “Health, Lifestyle, and Physical Fitness in Older Adults in Amazonas” (SEVAAI). PA was assessed using the Baecke Questionnaire, PF using the Senior Fitness Test, and GS using the 50-foot Walk Test. Mediation pathways were analyzed to test the possible mediating role of PF between specific PA domains (PA-total score, PA-housework, PA-sport, PA-leisure) and GS. Regarding PA-total, the analysis showed that high-performance GS was partially mediated in approximately 19% by better PF performance. Moreover, the PF could partially mediate the association between PA-sport and PA-leisure with GS, at levels of approximately 9% and 46%, respectively. An inverse relationship was observed between PA-housework (sedentary lifestyle) and GS. This association was partially mediated to an extent of approximately 9% by better PF performance. We conclude that PF plays a crucial role in mediating the association between PA and GS among vulnerable older adults.

## 1. Introduction

Physical function (PF) is strongly related to the human ability to perform activities related to daily living [1,2]. Especially in old age, adequate levels of PF are essential for physical and motor independence, which are necessary for an autonomous life [3]. With vulnerability, PF can decrease dramatically [4]. The most common manifestation of poor health status among the older population is represented by the loss of functioning [5]. Therefore, being an older person and having an impaired PF (e.g., with respect to cardiorespiratory performance, muscle strength, flexibility, speed, or balance) decreases the chance of having a good health condition, quality of life, and well-being [6]. Moreover, having a low level of physical function (PF), and pursuing sedentary behavior (SB) [7], which may be common in the older population [8], contributes to the individual having a low level of PA [9]. Thus, the combination of a low level of PF with a sedentary lifestyle impairs the mobility of older adults.

Mobility is defined as the human ability to move independently and safely in the environment [10]. In old age, mobility problems, including changes in gait speed (GS), are considered early indicators of physical health decline, a warning sign of functional disability [11]. Changes in mobility can negatively affect an older person’s autonomy [12], compromising their quality of life and well-being [13]. A low GS may result from endogenous changes caused by physiological aging, such as the loss of muscle strength, central/peripheral nervous systems, and proprioceptive feedback [14,15]. GS impairment is also strongly associated with a greater chance of falling [16]. Among older adults, falls represent a considerable risk to health and quality of life, and are responsible for injuries, days of hospitalization, and enhanced functional incapacity [17,18].

Concerning PA, its performance is classified into levels of intensity, including low intensity (e.g., daily activities such as cooking, cleaning, and working in the garden), moderate intensity and high intensity (e.g., physical exercise) [19]. Regardless of age group, having a low PA level represents an increased risk of developing chronic diseases [20] and a decreased chance of maintaining an autonomous life. Moreover, previous studies have associated low levels of PA with gait disturbances [17,19]. A useful strategy for combating low PA, which potentiates SB, is to increase PF levels [20,21]. In practice, this can be achieved through the regular performance of exercises focused on muscle strength, power, flexibility, cardiorespiratory endurance, postural control, and activities designed to enhance mobility [10,21].

The relationship between gait performance and age is inversely proportional [22]. Thus, the spatial and spatio-temporal parameters of gait tend to decrease [23]. Therefore, when aging is accompanied by low levels of PA and PF becomes insufficient, the performance of the GS automatically declines [24,25]. For this reason, comparatively sedentary older adults adopt a more cautious walking style than active ones, taking shorter and slower steps [26]. Therefore, in situations that require an increase in GS, older adults tend to increase cadence rather than stride length, while younger individuals do the opposite [27].

The relationships between aging, PA, PF, and gait parameters have been extensively studied [20,28,29]. However, the causes of the decrease in GS and changes in other gait patterns (e.g., cadence, stride time, step time, single support, double support, foot off, stride length, step length, time of stance, swing) are not entirely clear [7,30,31]. Furthermore, it is also unclear what weight each specific domain involved in the objective assessment of PA (e.g., PA-housework, PA-sport, PA-leisure) exerts on the GS of the cognitively normal and healthy older population. Thus, it is important to develop studies that expand our understanding of this relationship, and the findings may favor the creation of public health policy targets for specific domains of PA. Another critical point in the relationship between PA, PF, and GS, is better understanding the mechanisms of these variables in older adults residing in a given geographic area. In healthcare, geographic information plays an essential role in disease surveillance, management and analysis [32]. To the best of our knowledge, no study has addressed the mediating role of PF in the relationship between total PA and GS in the older population of Northern Brazil. Compared to other Brazilian states, the inhabitants of this region still live in conditions of extreme socioeconomic vulnerability [33,34]. According to the Brazilian Society of Geriatrics and Gerontology [35], this negatively affects the aging process of these citizens.

Thus, knowing that moderate to high levels of PF can improve GS performance [16,35], that there is a positive and direct relationship between PA and PF [24,36,37], and that high PF levels are essential to improving and/or maintaining adequate levels of GS [6,38], this study aimed to explore the mediating role of PF in the relationship between PA and GS in a large sample of older adults.

## 2. Materials and Methods

### 2.1. Study Design

This is an analytical cross-sectional observational study carried out in the Amazonas state, Northern Brazil. Participants took part in a research project entitled “Health, Lifestyle and Fitness in Adults and Elderly in Amazonas” (SEVAAI), carried out between 2016 and 2017. Participants were recruited through newspapers, churches, support centers, groups, or associations of older people in the municipalities of Manaus, Fonte Boa, and Apuí. The procedures followed the ethical principles, contained in Resolution 466/12 of the National Health Council of the Ministry of Health, evaluated and approved by the Human Research Ethics Committee of the University of the State of Amazonas (nº 1,599,258–CAAE: 56519616. 0000.5016).

### 2.2. Sample Size

For the sample size calculation, we used the G*Power [39]. A priori, using the F family tests, linear regression analysis, with two tested predictors from a total number of 5, indicated that to detect a small relation of *r* = 0.04, with a two-tailed alpha probability of 0.01 and a power of 0.99, the sample would need to comprise at minimum 691 individuals.

### 2.3. Participants and Eligibility

The inclusion criteria adopted were: (1) living in one of the three cities in Manaus mentioned above; (2) minimum age of 60 years; (3) able to walk independently and perform physical assessments; (4) present autonomy and independence to carry out activities of daily living; (5) no indication of serious health problems (medical contraindications for the practice of physical activity). An exclusion criterion was adopted: score < 15/30 on the Mini-Mental State Examination (MMSE) [40]. This criterion was considered the limit of inability to understand and follow the SEVAAI study protocol. All participants were informed about the investigation procedures and voluntarily signed an informed consent form. In the SEVAAI study, 701 people met the criteria and were included.

Figure 1 shows the flowchart with the study sample selection. In our mediation study, four participants from the original SEVAAI study database were excluded, one due to Parkinson’s disease (*n* = 1), and three due to Alzheimer’s disease (*n* = 3). Thus, the final sample consisted of 697 participants, totaling 267 men (71.4 ± 7.0 years) and 430 women (69.7 ± 6.6 years).

### 2.4. Data Collection

#### 2.4.1. Demographics and Clinical Data

The collection of information on gender, age, years of education, falls, medication, visual and hearing impairment, and blood pressure was obtained through self-report, obtained individually through face-to-face interviews using health questionnaire employed in the “FallProof!” Program [41]. The procedure was conducted by specially trained field team members.

#### 2.4.2. Anthropometry

Body mass index (BMI) was measured using an anthropometric scale and a Welmy^®^ stadiometer coupled with 0.1 cm and 0.1 kg [42], and calculated from weight and height (kg/m^2^).

#### 2.4.3. Cognitive Function

Mini-Mental State Examination (MMSE) was used to detect possible cases of dementia [43]. In the original SEVAAI study, a score of <15/30 points was assumed to be the disability threshold for participants to understand the protocols and follow all assessments.

#### 2.4.4. Physical Activity

The level of physical activity (PA) was measured using the Brazilian version of the Baecke Questionnaire for older adults [44], adapted from the original questionnaire by Voorrips et al. [45]. This instrument is divided into three sections based on lifestyle habits for the last 12 months: (1) household activities (PA-housework); (2) sports activities (PA-sport), only regular activities lasting at least one hour per week; and (3) free time activities (PA-leisure). In this study, the following scores were used: specific physical activity domains (PA-housework, PA-sport, PA-leisure), and the total score (PA-total), calculated from the mean scores of the three domains.

#### 2.4.5. Physical Function

PF was evaluated using the Senior Fitness Test (STF) [46]. For the present study, six physical function components were selected as physical fitness parameters: (1) lower body strength (quadriceps, glutes): after a signal, participants were asked to get up from a chair and then return to a fully seated position, repeating the task as many times as possible for 30 sec; (2) arm curl, to assess upper-body strength: after the signal, participants were instructed to flex and extend the elbow of the dominant hand, throughout the range of motion, lifting a weight (2.3 kg dumbbell for women, and 3.6 kg dumbbell for men) as many times as possible for 30 sec. The score was determined by the total number of repetitions performed in 30 sec; (3) flexibility in the lower body (sit-and-reach chair/cm): participants were asked to sit on the edge of a chair, with one leg bent and the other extended straight in front, keeping the heel on the floor, without bending the knee. In this position, the participants extended their hands in front of them, slowly sliding over the extended leg towards the feet. The score was the number of centimeters before the toes (lowest score) or reached beyond the toes (highest score); (4) flexibility in the upper body (back scratch/cm): participants placed one hand behind the same lateral shoulder with the forearm pronated, fingers extended, the other hand behind the back, fingers extended. The score was determined by the centimeters left for the middle fingers to touch those of the other hand (lower score), or centimeters overlapping each other (higher score); (5) agility/dynamic balance (8-foot up-and-go/s): participants were fully seated in a chair, hands on thighs and feet flat on the floor. After a signal, they got up from the chair, walked as quickly as possible (without running) around a cone placed 8 ft (2.44 m) in front of the chair, returning and sitting fully in the chair. The result was established by the time in seconds required to rise from a sitting position, walk and return to a sitting position; and (6) aerobic endurance (6-min walk test/m): after a signal, participants walked as fast as possible (without running) along a marked path, as many times as possible. The score was determined by the distance (meters) covered in the six-minute interval. A sum of the scores for all indicators provided by the SFT was used to calculate a continuous overall measure of the participants’ physical function (PF total = CST + ACT + CSAR + BST + FUG + MWT6).

#### 2.4.6. Gait

GS was assessed using the 30-Foot (9 m) Walk Test [41]. Participants were required to walk at their preferred speed. For each participant, three measurements were collected, and the best performance was considered in the analysis. A full description of the test administration instructions for the test is reported in Rose [41].

#### 2.4.7. Statistical Analysis

The main characteristics of the participants (i.e., sociodemographic, clinical, blood pressure, BMI and MMSE) are presented using descriptive statistics. These data are presented on the basis of the total number of individuals, and although our study does not focus on drawing group comparisons, these data are also presented in two groups according to the total score of the Baecke Questionnaire [44]. The calculation was performed based on the mean of the PA-total: PA-total < 2.57 points (low level), and PA-total ≥ 2.57 points (high level). From this, the main characteristics of the participants were compared using the Chi-square test (categorical variables) and the parametric Student’s t test for independent samples (continuous variables). In the descriptive statistics, only sociodemographic, clinical, blood pressure, BMI, and MMSE variables were included. Prior to the mediation analysis (the main objective of the investigation), the strength and direction of the association between the study variables (PA-total, PA-housework, PA-sport, PA-leisure, PF, and GS) were verified. The examination was performed by bivariate analyses, and the results were presented by Pearson’s correlation coefficients (*r*), considering the following interpretation: 0.1 = small, 0.3 = medium, and ≥0.5 = large [47].

Finally, we used the statistical mediation analysis method (Figure 2), making it possible to expand and qualify the understanding of how physical activity acts on GS influenced by the indirect effect of PF. A complete mediation would be observed if, with the inclusion of objectively measured PF (mediator variable), the size of association between the independent variable (PA-total, PA-housework, PA-sport, PA-leisure) and the dependent variable (GS) became non-significant, indicated by its confidence interval including zero [48]. A partial mediation would occur if the observed relationship between independent variable and dependent variable became weaker after the inclusion of objectively measured PF (mediator variable).

The effects represented by the regression coefficients in Equations (1) and (2) were estimated with a computational complement to the SPSS program using PROCESS v4.0: an analysis of model estimation developed by Hayes [49]. The coefficients a1 and b1 described in the equation (Figure 2) were calculated using least squares regression, as follows:M = *a*_0_ + *a*_1_ X + *r*
(1)
M = *b*_0_ + *c*^′^ X + *b*_1_ *M* + *r*(2)

The mediation hypothesis test was estimated using a confidence interval (95%) with bias correction and acceleration (BCa) by the Bootstrapping method with bias correction (5000 re-samplings). Thus, the indirect effect was considered significant when the confidence interval did not include zero [48]. The calculation of the proportion of the mediation effect was obtained as follows: subtraction 1 minus the result of the division between the direct effect and the total effect [49]. Furthermore, the results illustrated in the figures correspond to standardized parameters *β*.

## 3. Results

### 3.1. Main Characteristics of the Participants

The mean age of the group with low PA was 71.17 ± 7.28 years, while participants with high PA indicated a mean of 69.53 ± 6.31 years (Table 1). Regarding PA, except for the PA-housework domain (*p* = 0.024), the group with high PA exhibited better scores in the PA-sport and PA-leisure domains (*p* < 0.001). The high PA group members indicated better performance on functional tests, except for the 30 s chair stand test (CST) (*p* = 0.790). Regarding the GS test, members of the low PA group had a lower result than those of the high PA group (*p* < 0.001).

Regarding the descriptive analyses, Table 2 shows the levels of correlation between PA, PF and GS. A positive and small association was found between PA-housework and PA-sport (*r* = 0.245). A negative and small association was found between PA-housework and PA-leisure (*r* = −0.005). PA-leisure was not related to PA-housework. Higher scores on the PA-total were positively and weakly associated with higher scores on the PA-housework (*r* = 0.003), positively and at a medium level with the PA-sport (*r* = 0.476), in addition positively and at a large level with the PA-sport (*r* = 0.476), and PA-leisure (*r* = 0.870). A high level of performance in PA-total was negatively and weakly associated with PA-housework (*r* = −0.118), positively and weakly with PA-sport (*r* = 0.139), and with PA-leisure (*r* = 0.160). High values of GS were negatively and weakly associated with PA-housework (*r* = −0.215), positively and at a medium level with PA-sport (*r* = 0.515), positively and weakly with PA-leisure (*r* = 0.075), and with PA-total (*r* = 0.298). Finally, high performance in the GS indicated a positive and medium association with higher scores in the performance of PA-total (*r* = 0.302).

### 3.2. Mediation Analysis: PF in the Relationship between PA-Total and GS

Regarding the main objective of our study, Figure 3 presents the mediation model used to determine whether PF performance can mediate the effect of PA-total level on GS. The direct effect estimated by the model (*x* → *y*) showed a significant positive relationship between the highest level of PA-total and the highest GS, *β* = 0. 03; 95% CI (0.021–0.039), *t* = 6.64, *p* = 0.001. Path (a) = association between PA-total (*x*) with mediator Physical Function (*m*), Path (b) = association between mediator Physical Function (*m*) with Gait Speed (*y*), The total effect of the model (*x* → *y*) indicated a significant positive relationship between a high PA-total and high GS performance, *β* = 0.04; 95% CI (0.028–0.047), *t* = 8.14, *p* = 0.001, and a significant positive indirect effect was indicated between high PA-total and high GS performance, *β* = 0.01 (95% CI BCa = 0.0347–0.010). Thus, the proportion of the total effect of FA-total on GS mediated by PF was approximately 16%.

### 3.3. Mediation Analysis: PF in the Relationship between PA-Housework and GS

Figure 4 presents the estimate for the PA-housework specific domain. The direct effect estimated by the model (*x* → *y*) indicated a significant negative relationship between the highest level of PA-housework with low GS performance, *β* = −0.25; 95% CI (−0.3185–0.1847), *t* = −7.38, *p* = 0.000. Path (a) = association between PA-housework (*x*) with mediator Physical Function (*m*), Path (b) = association between mediator Physical Function (*m*) with Gait Speed (*y*). The total effect of the model (*x* → *y*) also showed a significant negative relationship between PA-housework and GS, *β* = −0.21; 95% CI (−0.2841–0.1430), *t* = −5.94, *p* = 0.001. In view of the indirect effect, a significant negative association was found between high PA-housework and low GS performance, *β* = −0.04 (95% CI BCa = 0.0127–0.0653). Thus, the proportion of the total effect of PA-housework on GS mediated by PF was approximately 9%.

### 3.4. Mediation Analysis: PF in the Relationship between PA-Sport and GS

In relation to the PA-sport specific domain (Figure 5), the direct effect estimated by the model (*x* → *y*) revealed a non-significant relationship between the highest level of PA-sport and the highest performance of GS, *β* = 0.12; 95% CI (0.1370–0.4816), *t* = −15.19, *p* = 0.132. Path (a) = association between PA-sport (*x*) with mediator Physical Function (*m*), Path (b) = association between mediator Physical Function (*m*) with Gait Speed (*y*), The total effect of the model (*x* → *y*) also showed a non-significant relationship between PA-sport and GS, *β* = 0.13; 95% CI (−0.1457–0.5142), *t* = 15.78, *p* = 0.086. On the other hand, the indirect effect indicated a significant positive association between high PA-sport and high GS performance, *β* = 0.01 (95% CI BCa = 0.0039–0.0133). Thus, the proportion of the total effect of PA-sport on GS mediated by PF was approximately 9%.

### 3.5. Mediation Analysis: PF in the Relationship between PA-Leisure and GS

For the PA-leisure specific domain (Figure 6), the direct effect estimated by the model (*x* → *y*) revealed a non-significant direct relationship between the highest level of PA-leisure and the highest GS, *β* = 0.04; 95% CI (0.0143–0.0249), *t* = 0.67, *p* = 0.496. Path (a) = association between PA-leisure (*x*) with mediator Physical Function (*m*), Path (b) = association between mediator Physical Function (*m*) with Gait Speed (*y*), The total effect of the model (*x* → *y*) also showed a positive and non-significant relationship between high PA-leisure and high GS performance, *β* = 0.02; 95% CI (−0.1457–0.5142), *t* = 1.91, *p* = 0.055. The indirect effect showed a positive significant relationship between high PA-leisure and high GS performance, *β* = 0.08 (95% CI BCa = 0.0037–0.0111). Thus, the proportion of the total effect of PA-leisure on PF-mediated GS was approximately 46%.

## 4. Discussion

This study aimed to explore the mediating role of PF in the relationship between PA-total and their specific domains with GS. Our findings showed that high levels of PA-total, PA-sport and PA-leisure were positively associated with better GS performance. In relation to total-PA, the analysis showed that a high GS was partially mediated by approximately 19% by better PF performance. Second, in relation to PA-sport, when PF was placed as a mediator, the direct and total effects of the path between *x*-*y* became non-significant. This means that PF partially mediated the association between PA-sport and GS in approximately 9%. Third, the inclusion of PF as a mediator showed that the direct and total pathway effects between PA-leisure and GS were not significant. This means that PF could partially mediate the association between PA-leisure and GS in approximately 46%.

The specific domain PA-housework was negatively associated with GS performance. Thus, a high PA-housework score was partially mediated by approximately 9% by better PF performance. The negative effect (direct and total) indicated by PA-housework on GS was mediated by a positive indirect effect of PF. This result suggests the importance of reducing SB levels in older adults and increasing their PF levels through physical exercise. In this way, it is possible to attenuate negative associations between PA and GS and, consequently, improve mobility, functionality, and quality of life [16]. SB is considered the main risk factor for noncommunicable diseases [50], responsible each year for the death of 41 million people, equivalent to 71% of all deaths in the world [51]. For this reason, the WHO has suggested guidelines for the older adult population entitled ‘Global Recommendations on Physical Activity for Health’ [52], as well as the American College of Sports Medicine [53]. The positions posted by both are similar, suggesting a minimum of 150 min of moderate-intensity aerobic activity per week, or at least 75 min of vigorous-intensity aerobic activity, or an equivalent combination. In addition, muscle-strengthening activities should be done on 2 or more days, and people with limited mobility should do balance exercises to prevent falls on 3 or more days. From this perspective, reaching the recommended levels of aerobic fitness, as well as a high physical condition can benefit the older adult population to improve several aspects associated with a good quality of life [54].

Age has an inversely proportional effect on PF [55]. Thus, an increase in PA levels may consequently generate an improvement in PF, presumably benefiting variables associated with mobility such as muscle strength, balance, and cardiovascular endurance [56], and notwithstanding, the GS itself [21]. Among sexagenarians, the annual muscle loss varies from 1 to 2%, reaching 3.4% after age 75 [4]. Furthermore, the loss of muscle strength is different between the sexes and between the lower and upper limbs. On the other hand, it is known that physiological changes resulting from aging, as well as functional limitations associated with mobility, can be mitigated or prevented through physical exercise, increasing daily/weekly PF levels [20].

Our evidence that a higher level of PF is associated with a higher GS is in accordance with previous investigations. In a cross-sectional study (*n* = 1352; 68.6 ± 7.5 years), the association of PA-total with GS was observed for those aged ≥ 75 years [57]. Therefore, moderate and vigorous PA levels were related to lower-body physical performance (GS and the task of rising from a chair) and handgrip strength. Regarding sedentary time, the authors identified inverse associations with lower-body physical performance [57], which can negatively affect GS. This result is in agreement with our finding that the performance of the GS tends to be slower, when the level of PA-housework was also low.

At advanced age, the interdependence between PA, PF, and GS is considerable and decisive for the individual’s autonomy. High levels of PA [58,59] and high GS performance are essential for safely performing instrumental activities of daily living [60]. A GS performance of 1 m/s is considered the ideal minimum threshold for an older adult to have a more stable gait pattern [23]. Gait is also a strong indicator of health and quality of life [10]. On the other hand, a low GS may be associated with poor cognitive performance [61], as well as leg muscle weakness [62], and consequently, these factors together may increase the risk of falls [60,63,64]. Moreover, slow GS values can also be indicative of future cognitive impairment [65]. For this reason, GS plays a role as a marker of cognitive decline. A longitudinal study indicated a negative relationship between PA and cognitive decline during vulnerable aging [66]. For this reason, a suggested strategy to reverse/mitigate this relationship is to promote PA levels. Consequently, there is a greater chance that cognitive functions will remain adequate [67]. Another population-based longitudinal investigation revealed an increased risk of morbidity [68], and mortality for older adults who showed a rapid decline in GS [69,70]. In a population-based cohort study conducted with older adult Brazilians (>60 years; *n* = 332) to assess the relationship between PA and SB with mortality revealed SB as a potential risk for mortality [71]. Moreover, the analysis pointed to SB as a barrier for older adults to present moderate to vigorous PA levels.

It is worth noting that PA and/or SB levels may be strongly associated with socioeconomic factors. In a cross-sectional study that used data from a national household-based survey carried out by the Brazilian Ministry of Health (*n* = 60,202) to examine the population’s health status, lifestyle, and chronic diseases [72], no differences were found regarding gender. Still, differences were found in ethnicity (being older adult black or white), weekly practice of physical exercise, income, and years of education. Having a low income and few years of education can prevent older adults from practicing physical exercises, as well as demonstrate low involvement in leisure activities. In another population-based study in in southeast Brazil (*n* = 621; 70.8 ± 8.1 years), the prevalence of SB was 70.1% [73]. Moreover, being male and over 80 years, having few years of education, low functional capacity, smoking, and not having private health insurance were associated with SB. Although it was not our objective, we partially corroborate these results. Indeed, in our study, members of the high PA group indicated 3.38 (48%) more years of education than those in the low PA group.

The present study’s findings confirmed the outcomes of previous studies, suggesting that high levels of PA are essential for promoting and/or preventing the decline of functions involved in PF, which plays a key role in healthy aging [74], including preserved GS. The ability to walk depends on the good function and interaction of a set of systems (e.g., musculoskeletal, visual, central, and peripheral nervous) and cardiorespiratory fitness [68]. Highlighting the role of gait quality in the life of the older adults, it can be said that small changes in GS (an increase of 0.1 m/s) were related to an increase in the predicted survival of 10 years, with a variation of 19% to 87% for men, and 35% to 91% for women [68]. In this context, we reiterate that to reduce or prevent loss of speed during gait, it is essential to increase moderate/vigorous PA frequency and duration throughout the week [75].

The cross-sectional design used in this study is a limitation, since it does not allow conclusions about the cause-and-effect relationship between PA and GS when mediated by the role of PF Moreover, cross-sectional data cannot support causal inferences, but mediation analysis is considered a robust approach, therefore making it possible to infer potential causalities. A strong point of this investigation is the more in-depth results of the relationship between the three specific domains of PA with GS when mediated by PF. Another strong point is the inclusion of a large representative sample from a defined geographic area [32] (the municipalities of Manaus, Fonte Boa, and Apuí). Thus, this is the first mediation study carried out with community-dwelling older adults from the north of Brazil. The information about PA was self-reported, and therefore, possible bias derived from the data collection must be considered. However, to reduce the risk of bias, the interviews were conducted in person by trained researchers. It is suggested that future investigations focusing on the older population might explore the association between specific domains of PA and GS, investigating the mediating role of PF both in older adults in other regions of Brazil and in other countries to compare and extend our findings. Moreover, we suggest the inclusion of longitudinal monitoring.

## 5. Conclusions

We conclude that PF is crucial in mediating the association between PA and GS among vulnerable older adults. Our findings also highlight the importance of high levels of PA, particularly PA-sport and PA-leisure, combined with high levels of PF, for older adults’ GS to maintain enough mobility to guarantee their autonomy. Therefore, investing in greater engagement of the older population in sports and leisure activities can presumably be an effective, simple, and inexpensive strategy to promote GS, quality of life, and well-being. Finally, this study contributes to a better understanding of the interrelationship between PA, PF, and GS in vulnerable aging, helping to formulate helpful strategies for public health guidance.

## Figures and Tables

**Figure 1 ijerph-19-12581-f001:**
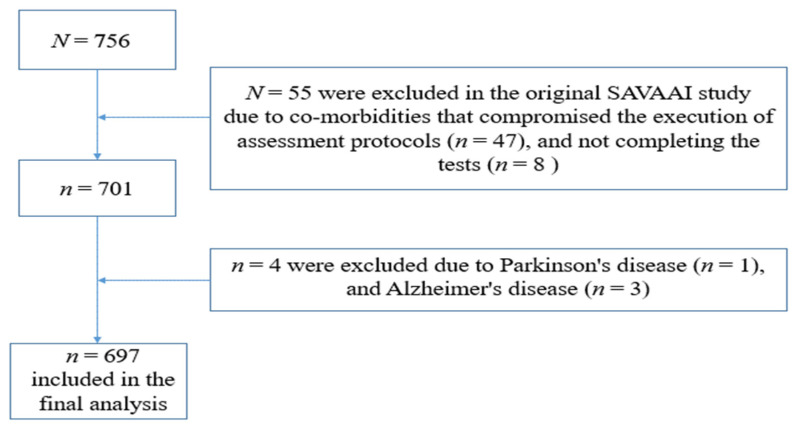
Flowchart of study sample.

**Figure 2 ijerph-19-12581-f002:**
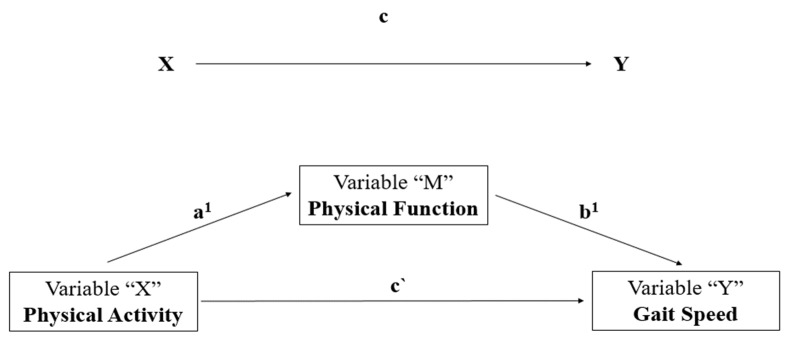
Linear mediation relationships to examine the mediating role of PF in the relationship between PA and GS. Path (a) = association between PA-Activity (X) with mediator (M) Physical Function (Y), Path (b) = association between mediator (M) Physical Function with Gait Speed (Y), Path c’ = direct effect (X-Y).

**Figure 3 ijerph-19-12581-f003:**
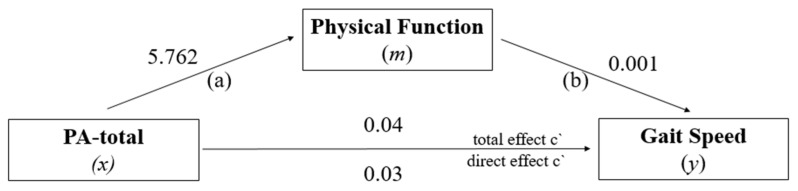
PF model as a mediator of the effect of PA-total on GS.

**Figure 4 ijerph-19-12581-f004:**
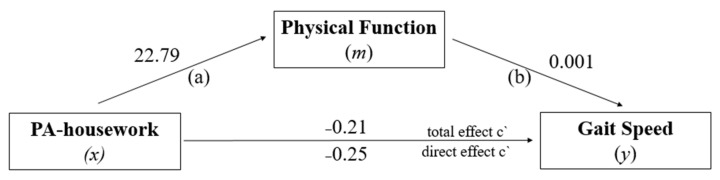
PF model as a mediator of the effect of PA-housework on GS.

**Figure 5 ijerph-19-12581-f005:**
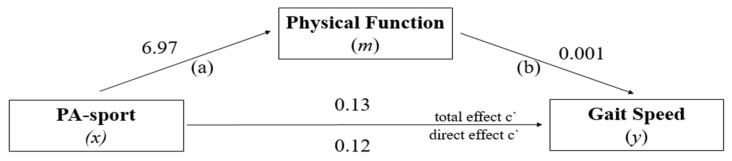
PF model as a mediator of the effect of PA-sport on GS.

**Figure 6 ijerph-19-12581-f006:**
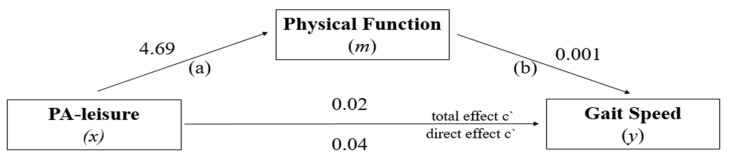
PF model as a mediator of the effect of PA-leisure on GS.

**Table 1 ijerph-19-12581-t001:** Main characteristics of participants, according to the level of physical activity.

Variable	Low PA Mean (SD)	High PA Mean (SD)	Total Mean (SD)	*p*-Value
	(***n* = 348**)	(***n* = 349**)	(***n* = 697**)	
Age (years)	71.17 ± 7.28	69.53 ± 6.31	70.35 ± 6.36	0.007
Sex (*n*) (%)				
women	196 (56.3)	234 (67.0)	430 (61.7)	0.004
men	152 (43.7)	115 (33.0)	267 (38.3)	
BMI (k/m^2^)	27.88 ± 5.00	25.30 ± 3.86	28.21 ± 4.93	0.059
Education (years)	3.65 ± 4.72	7.03 ± 7.80	5.35 ± 5.55	<0.001
MMSE (*n*)	23.50 ± 4.40	25.30 ± 3.86	24.41 ± 4.23	<0.001
Falls n (%)	121 (34.8)	106 (30.4)	227 (32.6)	0.216
Medication (*n*)	1.83 ± 1.83	1.95 ± 1.74	1.89 ± 1.79	0.255
Comorbidities (*n*) (%)				
Hypertension	201 (57.8)	194 (55.6)	395 (56.7)	0.563
Visual impairment	289 (83.0)	293 (84.0)	582 (83.5)	0.012
Hearing problems	85 (24.4)	95 (27.2)	180 (25.8)	0.555
Physical activity				
PA-housework (*n*)	2.87 ± 0.44	2.78 ± 0.45	2.82 ± 0.47	0.024
PA-sport (*n*)	2.00 ± 0.40	2.35 ± 0.62	2.18 ± 0.55	<0.001
PA-leisure (*n*)	2.70 ± 0.51	2.93 ± 0.55	2.71 ± 0.54	<0.001
PA-total (*n*)	2.52 ± 0.45	2.68 ± 1.53	2.57 ± 0.52	<0.001
Physical function				
CST (*n*)	11.71 ± 3.41	11.74 ± 3.02	11.73 ± 3.22	0.790
ACT (*n*)	13.92 ± 4.41	12.36 ± 3.91	13.14 ± 4.24	<0.001
CSAR (cm)	1.97 ± 8.02	3.77 ± 10.70	2.87 ± 9.50	0.021
BST (cm)	−12.80 ± 12.90	−6.87 ± 10.61	−9.84 ± 12.17	<0.001
FUG (seg.)	6.71 ± 2.45	5.88 ± 1.20	6.30 ± 1.98	<0.001
MWT 6 (m)	400.47 ± 89.90	438.98 ± 80.16	419.75 ± 87.25	<0.001
PF total (score)	422.75 ± 94.84	465.86 ± 86.11	444.40 ± 93.04	<0.001
Gait speed (m/s)	1.17 ± 0.36	1.52 ± 0.49	1.35 ± 0.46	<0.001

BMI: body mass index; MMSE: Mini Mental State Examination; CST: 30 s chair stand test; ACT: 30 s arm curl test; CSAR: chair sit-and-reach test; BST: back scratch test; FUG: foot up-and go test; MWT6: 6-min walk test; *p* < 0.05; CI, confidence interval.

**Table 2 ijerph-19-12581-t002:** Associations between analyzed variables.

Variable	PA-Housework	PA-Sport	PA-Leisure	PA-Total	PF
PA-sport	0.245 ***				
PA-leisure	−0.005 ^ns^	0.000 ^ns^			
PA-total	0.003 ^ns^	0.476 ***	0.870 ***		
PF	0.118 **	0.139 ***	0.160 ***	0.226 ***	
Gait speed	−0.215 ***	0.515 ***	0.075 *	0.298 ***	0.302 ***

PA = physical activity; PF = physical function; * *p* < 0.05; ** *p* < 0.01; *** *p* < 0.001; ^ns^ = non-significant, *p* > 0.05.

## Data Availability

The data presented in this study are available upon request from the corresponding author.

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
