# Peer review of "The Role of Physical Function in the Association between Physical Activity and Gait Speed in Older Adults: A Mediation Analysis"

_ijerph, 2022, doi:10.3390/ijerph191912581_

Round 1
Reviewer 1 Report
Thanks for correcting the minor issues reported previously.
Author Response
Dear Reviewer, we thank you for your attention and contributions to the improvement of the manuscript.
Sincerely!
The authors
Reviewer 2 Report
The paper has a relevant subject. The type of analysis used is appropriate for the topic, and the conclusions are clear. I only ask that you explain in more detail the origin of the percentages in the results of each mediation analysis. In topics 3.2 to 3.5 contains paragraphs containing the values of images 2 to 5, but the percentage I can not identify where comes from. On line 153, the term “FallProof!” I believe it is an error. Congratulations.
Author Response
Dear Reviewer, we thank you for your contributions. All requests were worked on in the manuscript (marking in yellow color). If necessary, we are available for future corrections!
1. I only ask that you explain in more detail the origin of the percentages in the results of each mediation analysis. Topics 3.2 to 3.5 contain paragraphs containing the values of images 2 to 5, but the percentage I can't identify where it comes from.
Replay:
Dear Reviewer, the calculation of the mediation ratio was described at the end of section "2.4.7 Statistical analysis" (page 6, lines 283-285).
2. On line 153, the term “FallProof!” I believe it's an error.
Reply:
Dear Reviewer, the term "FallProof" is correct. You can review this in reference [41]: Rose, D.J. Fallproof!: The Comprehensive Balance and Mobility Training Program; 2nd ed.; 2010; ISBN 978-0-7360-6747-8.

Reviewer 3 Report
Introduction
This section is very complete and puts the reader in the background. Likewise, the references used are recent and have scientific relevance.
Methods
This section needs a series of improvements:
• Authors must add a flow diagram of the participants (information appears in the text, but must appear in a figure).
• The calculation of the sample must be added in a new subsection with its corresponding reference.
• The title of section 2.2.3. Cognitive function. Please, check that it appears with a different format from the rest of the manuscript (it seems that it has been made with change control and they have not been accepted).
Discussion
This section adequately reflects the scientific evidence related to this study and correctly compares its results with those of other authors. Please review lines from 351 to 363 and from 426 to 435 that appear in a different format than the rest of the manuscript (it seems that it has been made with change control and they have not been accepted).
Conclusion
This section reflects the conclusions of the study correctly. Please check that it appears with a different format from the rest of the manuscript (it seems that it has been made with change control and they have not been accepted).
Author Response
Dear Reviewer, we thank you for your contributions. All requests were worked on in the manuscript (marking in yellow color). If necessary, we are available for future corrections!
Methods
1. Authors must add a flowchart of participants (the information appears in the text, but must appear in a figure).
Reply:
Dear Reviewer, a flowchart was created and included (Figure 1) (page 4, line 163-178)
2 The sample calculation must be added in a new subsection with its corresponding reference.
Replay:
Dear Reviewer, the requested sample calculation has been included in the section "2.2 Sample size" (page 3, lines 133-137).
3 The title of section 2.2.3. Cognitive function. Please check if it appears in a different format from the rest of the manuscript (it looks like it was made with control of changes and they were not accepted).
Reply:
Dear Reviewer, the formatting has been revised!
Discussion
1. Please review lines 351 to 363 and 426 to 435 that appear in a different format from the rest of the manuscript (it looks like it was done with control of changes and they were not accepted).
Reply:
Dear Reviewer, the formatting has been revised!
Conclusion
1. Please check if it appears in a different format from the rest of the manuscript (it looks like it was made with control of changes and they were not accepted).
Reply:
Dear Reviewer, the formatting has also been reviewed!

This manuscript is a resubmission of an earlier submission. The following is a list of the peer review reports and author responses from that submission.
Round 1
Reviewer 1 Report
Dear authors,
This is an excellent paper with significant socioeconomical impact. The key strengths are: your sample size and its geographical origin, the range of tests employed and the innovative statistical analysis. Your manuscript is professionally written with attention to detail. I believe that the findings can help future research but also policy and physical activity practice.
From a biomechanical perspective (which is my specialist area) I could make several comments and suggest additional checks and balances. However, the objective and the nature of this paper is somewhat different to a biomechanical study on gait analysis in the elderly. Therefore, I am content with the manuscript as it is, and I am recommending its publication.
Just please do a full spell check and correct the syntax issues, I spotted, below:
Lines 66-68: incorrect syntax – sentence does not make full sense.
Lines 68-69: The sentence is incomplete.
Lines 370-373: Expression needs improvement. Also, did you mean “adult” and not “adduct”.
Author Response
Dear Reviewer, we are grateful for all the comments, and are available for future clarifications and/or corrections.
* Changes were made in the text using Microsoft Word's built-in track changes function.
1. Lines 66-68: incorrect syntax – sentence does not make full sense.
2. Lines 68-69: The sentence is incomplete.
Reply
Dear Reviewer, thank you for your observation, the sentences have been corrected (page 2, lines 66-69).
3. Lines 370-373: Expression needs improvement. Also, did you mean “adult” and not “adduct”
Reply
Dear Reviewer, thank you for your observation, the expressions have been corrected (page 10, lines 417-418).

Reviewer 2 Report
Dear authors,
This paper has its merits, as it investigates important aspects of health in a largely understudied group of older people. However, there are numerous issues, mainly that the conclusion that high levels of PA with high levels of PF is crucial to improve and maintain GS among vulnerable older adults, from an analysis of cross-sectional data. I do not disagree that PA and therefore PF is important for gait speed, but it is not possible to draw these conclusions from crossectional data. Furthermore, the language needs a revision, as almost all sentences can be made significantly shorter, I give some examples below. Also all of the citations must be revised, as there are multiple occasions of erronous citations.
Introduction
line 60: The authors state that PF is related to ADL, where "related to" implies a statistical association but the reference [1] is a qualitative study.
line 68-69: "All this enchances the establishment of a circle of unfavourable factors for mobility". This is an example of how almost all sentences in this paper can be made much shorter: this sentence can be replaced with: "All this impairs mobility". This is in itself not precise enough in my opinion but can be fixed with "All this impairs mobility in older persons"
line 91-92 "Indeed, while the temporal parameters of life increase..." What does this mean? I could not find why the authors cited ref #22 for this sentence.
line 99: "However, the causes of decreased GS and changes in its quality standards are not entirely clear" What are "quality standards?"
Method
line 123: "this research" unclear what this means? The SEVAAI research project or this particular study?
line 132: "For the present study, four participants were excluded due to Parkinson's disease (n=1)" One or four participants exluded due to PD?
line 135-140: The authors state the criteria for the SEVAAI project and that 701 were included. They then state exclusion due to PD and AD, and a final sample of 697. After that they state inclusion criteria regarding age, ADL-independence, ability to perform assessments, no medical contraindication for physical activity, and exclusion critera of < 15 on the MMSE. With those criteria i would suspect that some participants would be excluded, but the final sample is still 697 to my understanding. The final sample size must be made clear, I suggest describing the SEVAAI first, with criteria and study sample, then a new paragraph with criteria for this particular study, and with the final sample stated last.
line 155-156: "Strength in the lower part" Is this refering to lower extremities, i.e. legs, lower back, or both? please specify.
line 157: "strength in the upper body (bar thread)" What is a "bar thread", a kind of exercise equipment? The authors should state which movement is performed, not the equipment used as for example a barbell can be used in a wide variety of movements.
line 160: abbreviation "SFT" was is not established previously.
line 166-167: "Participants were required to walk a distance of 50 feet at the preferred speed." Should it be "at their preferred speed"?
line 186-188: "A complete mediation would be observed if the inclusion of objectively measured PF (mediator variable) reduced to zero the association observed between independent variable (PA-total, PA-housework" What would be reduced to zero? the p-value, correlation coefficient or no overlapping confidence intervals?
Results
line 211: "The high PA group attested..." consider using another word, as attested can imply that the participants only declared better performance.
line 213: "Members of the group with and high PA also indicated superior performance..." what does "with and high" mean? Also, that the group indicated superior performance is unclear, did they have better performance or not? in the latter case, the authors should also state that the group with higher PA had superior performance compared to the low PA group.
line 214: Please be concise with wording as to not confuse the readers. Exam could mean a written exam. The word "tests" usually works well.
Page 5, table 1:
Hypertension, vision and hearing impairment is only mentioned in this table, the method section must describe how these are diagnosed (cutoffs systolic/diastolic bp vision and hearing impairment assessment). The authors must also describe if these are used as adjustments in any analyses or only used as descriptive statistics.
Page 5 table 1, the row for male, column "low PA": 47.3 %: these are neither column% or row%, I guess it should be the former, but at 43.7%. Also, usage of male and female is uncommon, please consider "men, women".
Mediation analys presentations: The authors present beta-values with 2, 3 and 4 decimals, please revise to use the same number of decimals.
line 250: "Non-standard parameters." please explain what this means.
line 256: a P-value can never be exactly zero, use the "<" sign if the p value is very low and you want to avoid too many decimals
line 258-259: "a significant negative association was found between high PA-housework and low GS performance,β = 0.038", negative associations have a negative b-value, but this one have not.
line 278: "and the highest GS, β = .036;" is this the beta value of 0.03 in fig 5? a rounding error?
line 280: beta value of 0.017, is this the 0.01 in fig 5? a rounding error?
Discussion:
The authors does not have a result summary, the structure is not good, different terms are used interchangeably (e.g. GS and gait, SB and physical inactivity)
-They state that e.g. "PF could fully mediate... by approximately 9%). It seems contradictory that something can fully mediate but still at only 9%. The authors should explain in more clear terms what these results mean.
line 293: "The premise" Do the author mean hypothesis? In that case, no such hypothesis was stated previously.
line 294-296: high GS was partially mediated (19%) by better PF performance, and when PF was placed as a mediator, no effect was present. I dont understand, wasn't PF the mediator in the first sentence, but then the authors states "when PF was placed as a mediator"? This must be made clearer.
line 311, ref 44: this is a reference to the 2010 guidelines, but the WHO released updated guidelines in 2020 that should be cited instead.
line 322: "studies" implies that the authors cited several studies, but I see only one.
line 328: "physical inactivty measured by PA housework", are the authors saying that housework is inactivity? If so, there are plenty of studies investigating MET values for common housework that the authors should read and revise this sentence.
line 329-330: This is just describing the Beacke questionnaire and therefore should be placed in the method section.
line 331-332: "how many people are benefited" what does this mean?
line 333: "All this reflects the mobility." Mobility of what, or who?
line 335-336: The authors state that high levels of PA and GS are essential to perform ADL safely, but they cite two cross-sectional studies. Crossectional studies are unable to provide evidence of causality. I do not question the premise that PA and GS are associated, and likely also in a causal way, but the authors should be able to find RCTs or at least longitudinal studies that support this claim.
line 338: "to present qualitatively mobility capacity" unclear what this means
line 338-339: "Gait is also a strong indicator of health and quality of life [9]. In this context, a poor GS is highly associated with an increased risk of falls [48,49,50]." Why do the authors use "In this context..." They mention health and QoL in the sentence before, which doesnt really translate to risk of falls.
line 359: "southeast region" does that mean southeast region of Brazil? Then it can be shortened to "in south-east Brazil".
line 371: "adduct", should it be adult?
line 389-391: "The combination of endogenous factors related to vulnerable physiological aging [50,61] with low PA levels contributes to a low PF level, potentiating a vicious circle capable of progressively or abruptly reducing GS." This sentence is long and hard to follow, and a conclusion usually doesnt involve any citations as it should summarize the conclusions of the present study, not other studies.
line 393-394: "break or at least maintain the mobility" what does it mean to break the mobility?
line 401-402: "On the other hand, it is worth noting that health promotion also depends on creating specific spaces for older adults to practise physical exercises and investments in the training of professionals to supervise physical exercises." Why is this mentioned? This study has not investigated health promotion via specific spaces or educating professionals.
Author Response
Dear Reviewer, we are grateful for all the comments, and are available for future clarifications and/or corrections.
Changes were made in the text using Microsoft Word's built-in track changes function.
ntroduction
line 60: The authors state that PF is related to ADL, where "related to" implies a statistical association but the reference [1] is a qualitative study.
Reply
Dear Reviewer, we have replaced the reference with two review studies and meta-analysis (page 2, lines 59-60).
line 68-69: "All this enhances the establishment of a circle of unfavorable factors for mobility". This is an example of how almost all sentences in this paper can be made much shorter: this sentence can be replaced with: "All this impairs mobility". This is in itself not precise enough in my opinion but can be fixed with "All this impairs mobility in older persons"
Reply
Dear Reviewer, we appreciate your suggestion, the sentence has been adjusted (page 2, lines 68-69).
line 91-92 "Indeed, while the temporal parameters of life increase..." What does this mean? I could not find why the authors cited ref #22 for this sentence.
Reply
Dear Reviewer, the sentences have been corrected, as well as the references updated with longitudinal studies (page 2, lines 91-93).
line 99: "However, the causes of decreased GS and changes in its quality standards are not entirely clear" What are "quality standards?"
Reply
Dear Reviewer, the expression "quality standards" has been replaced by "gait standard". Moreover, clear examples of "gait patterns" have been included (page 2, lines 98-100)
Method
1) line 123: "this research" unclear what this means? The SEVAAI research project or this particular study?
Reply
Dear Reviewer, the sentence has been corrected (page 3, line 123).
2) line 132: "For the present study, four participants were excluded due to Parkinson's disease (n=1)" One or four participants excluded due to PD?
Reply
Dear Reviewer, the sentence has been corrected (page 3, line 139-143).
3) line 135-140: The authors state the criteria for the SEVAAI project and that 701 were included. They then state exclusion due to PD and AD, and a final sample of 697. After that they state inclusion criteria regarding age, ADL-independence, ability to perform assessments, no medical contraindication for physical activity, and exclusion critera of < 15 on the MMSE. With those criteria i would suspect that some participants would be excluded, but the final sample is still 697 to my understanding. The final sample size must be made clear, I suggest describing the SEVAAI first, with criteria and study sample, then a new paragraph with criteria for this particular study, and with the final sample stated last.
Reply
Dear Reviewer, Section 2.1. Study design and participants has been reorganized (page 3, line 121-143)
4) line 155-156: "Strength in the lower part" Is this refering to lower extremities, i.e. legs, lower back, or both? please specify.
Reply
Dear Reviewer, we corrected the sentence using the expression "lower body strength" which is referred to in the Senior Fitness Test manual. We also clarify that the test is only intended to assess the strength of the leg muscles (quadriceps, glutes), (page 4, line 177-178).
5) line 157: "strength in the upper body (bar thread)" What is a "bar thread", a kind of exercise equipment? The authors should state which movement is performed, not the equipment used as for example a barbell can be used in a wide variety of movements.
Reply
Dear Reviewer, we replace the bar thread for a 2.3kg dumbbell for women and a 3.6kg dumbbell for men. In section 2.2.5, Physical Function, we explain better the movement in each physical fitness function test (please see page 4, from line 180-184).
6) line 160: abbreviation "SFT" was is not established previously.
Reply
Dear Reviewer, "STF" abbreviation for Senior Fitness Test has been introduced (page 4, line 176)
7) line 166-167: "Participants were required to walk a distance of 50 feet at the preferred speed." Should it be "at their preferred speed"?
Reply
Dear Reviewer, thanks for the observation, we corrected the test distance as well as its execution mode (page 4, line 207-208)
8) line 186-188: "A complete mediation would be observed if the inclusion of objectively measured PF (mediator variable) reduced to zero the association observed between independent variable (PA-total, PA-housework" What would be reduced to zero? the p-value, correlation coefficient or no overlapping confidence intervals?
Reply
Dear reviewer, for better understanding the sentences have been revised and rewritten (page 5, lines 230-234). There is also an explanation on page 255, followed by a reference: Preacher, K.J.; Rucker, D.D.; Hayes, A.F. Addressing Moderate Mediation Hypotheses: Theory, Methods and Prescriptions. Multivariate Behavior. Res. 2007, 42, 185-227.
Results
1) line 211: "The high PA group attested..." consider using another word, as attested can imply that the participants only declared better performance.
Reply
Dear Reviewer, the word "attested" was changed to "indicated" (page 6, line 257)
2) line 213: "Members of the group with and high PA also indicated superior performance..." what does "with and high" mean? Also, that the group indicated superior performance is unclear, did they have better performance or not? in the latter case, the authors should also state that the group with higher PA had superior performance compared to the low PA group.
Reply
Dear Reviewer, the sentence has been corrected (page 6, line 258-260).
3) line 214: Please be concise with wording as to not confuse the readers. Exam could mean a written exam. The word "tests" usually works well.
Reply
Dear Reviewer, as requested, the word exam has been replaced throughout the text with test (page 6, line 260).
4) Page 5, table 1:
Hypertension, vision and hearing impairment is only mentioned in this table, the method section must describe how these are diagnosed (cutoffs systolic/diastolic bp vision and hearing impairment assessment). The authors must also describe if these are used as adjustments in any analyses or only used as descriptive statistics.
Reply
Dear Reviewer,
Information on collecting data concerning vision and addiction problems as well as blood pressure has been included in the section 2.2. We used all the information to better describe this sample (i.e., gender, age, years of education, falls, medication, visual and hearing impairment, and blood pressure) based on a health questionnaire employed in the FallProof! Program (Rose 2010) “Data collection (page 3, from line 147);
- a) Regarding blood pressure, the question was: “Have you ever been diagnosed as having high blood pressure? In this study, we didn’t analyze de scores of diastolic and systolic blood pressure.
- b) The explanation to readers that vision, hearing and blood pressure problems were only considered in descriptive statistics has been included in the Statistical analysis section (page 4, line 213)
5) Page 5 table 1, the row for male, column "low PA": 47.3 %: these are neither column% or row%, I guess it should be the former, but at 43.7%. Also, usage of male and female is uncommon, please consider "men, women".
Reply
Dear Reviewer, thanks for the observation. The percentage and denominations (men, women) were corrected in Table 1.
6) Mediation analysis presentations: The authors present beta-values with 2, 3 and 4 decimals, please revise to use the same number of decimals.
Reply
Dear Reviewer, we have adjusted all "beta-values" results to two decimals.
7) line 250: "Non-standard parameters." please explain what this means.
Reply
Dear Reviewer, an explanation of the use of parameters has been included at the end of the Statistical analysis section (página 5, linhas 250-251). We use beta values (β), therefore we present standardized parameters compared to CI intervals.
8) line 256: a P-value can never be exactly zero, use the "<" sign if the p value is very low and you want to avoid too many decimals
Reply
Dear Reviewer, we have adjusted the p-value as requested (section 3.4. Mediation analysis)
9) line 258-259: "a significant negative association was found between high PA-housework and low GS performance,β = 0.038", negative associations have a negative b-value, but this one have not.
Reply
Dear Reviewer, we corrected the result by adding the negative symbol. We also rounded the value from 0.038 to 0.04 (section 3.4, PA-housework)
10) line 278: "and the highest GS, β = .036;" is this the beta value of 0.03 in fig 5? a rounding error?
Reply
Dear Reviewer, the issue regarding the beta-value in text and Figure 5 has been fixed.
11) line 280: beta value of 0.017, is this the 0.01 in fig 5? a rounding error?
Reply
Dear Reviewer, the issue regarding the beta-value in text and Figure 5 has been fixed
Discussion:
1) The authors does not have a result summary, the structure is not good, different terms are used interchangeably (e.g., GS and gait, SB and physical inactivity)
Reply
Dear Reviewer, the first paragraph of the Discussion section has been restructured. Furthermore, the terms gait and physical inactivity were adjusted throughout the text for GS and SB, respectively.
2) They state that e.g. "PF could fully mediate... by approximately 9%). It seems contradictory that something can fully mediate but still at only 9%. The authors should explain in more clear terms what these results mean.
Reply
Dear Reviewer, we agree with your suggestion. Therefore, we have corrected the term fully to partially throughout the text.
3) line 293: "The premise" Do the author mean hypothesis? In that case, no such hypothesis was stated previously.
Reply
Dear Reviewer, the sentence has been deleted, and we have also restructured the first paragraph of the Discussion section (page 9, 335-350)
4) line 294-296: high GS was partially mediated (19%) by better PF performance, and when PF was placed as a mediator, no effect was present. I dont understand, wasn't PF the mediator in the first sentence, but then the authors states "when PF was placed as a mediator"? This must be made clearer.
Reply
Dear Reviewer, we have corrected the information in the first paragraph (page 9, line 338-340).
5) line 311, ref 44: this is a reference to the 2010 guidelines, but the WHO released updated guidelines in 2020 that should be cited instead.
Reply
Dear Reviewer, we have updated the reference [48].
6) line 322: "studies" implies that the authors cited several studies, but I see only one.
Reply
Dear Reviewer, we have corrected the sentence, which should remain singular (page 9, line 371).
7) line 328: "physical inactivity measured by PA housework", are the authors saying that housework is inactivity? If so, there are plenty of studies investigating MET values for common housework that the authors should read and revise this sentence.
Reply
Dear Reviewer, we corrected the sentence (page 9, lines 376-378).
8) line 329-330: This is just describing the Beacke questionnaire and therefore should be placed in the method section.
Reply
Dear Reviewer, we have corrected the sentence, the detailed information from the Baecke Questionnaire has been removed, and adjusted in the Physical activity section (page 4, from line 167).
9) line 331-332: "how many people are benefited" what does this mean?
Reply
Dear Reviewer, as clarified, this passage has been removed from the text.
10) line 333: "All this reflects the mobility." Mobility of what, or who?
Reply
Dear Reviewer, as clarified, this passage has been removed from the text.
11) line 335-336: The authors state that high levels of PA and GS are essential to perform ADL safely, but they cite two cross-sectional studies. Crossectional studies are unable to provide evidence of causality. I do not question the premise that PA and GS are associated, and likely also in a causal way, but the authors should be able to find RCTs or at least longitudinal studies that support this claim.
Reply
Dear Reviewer, as requested, we have included two systematic review studies and one longitudinal study (page 9, lines 380-381)
12) line 338: "to present qualitatively mobility capacity" unclear what this means.
Reply
Dear Reviewer, the phrase has been corrected, using the expression more stable gait pattern (page 9, lines 383)
13) line 338-339: "Gait is also a strong indicator of health and quality of life [9]. In this context, a poor GS is highly associated with an increased risk of falls [48,49,50]." Why do the authors use "In this context..." They mention health and QoL in the sentence before, which doesnt really translate to risk of falls.
Reply
Dear Reviewer, the meaning of the sentence has been corrected (page 9, lines 383-386)
14) line 359: "southeast region" does that mean southeast region of Brazil? Then it can be shortened to "in south-east Brazil".
Reply
Dear Reviewer, thank you, the suggestion was followed (page 10, line 406)
15) line 371: "adduct", should it be adult?
Reply
Dear Reviewer, the sentence has been corrected (page 10, line 418).
16) line 389-391: "The combination of endogenous factors related to vulnerable physiological aging [50,61] with low PA levels contributes to a low PF level, potentiating a vicious circle capable of progressively or abruptly reducing GS." This sentence is long and hard to follow, and a conclusion usually doesnt involve any citations as it should summarize the conclusions of the present study, not other studies.
Reply
Dear Reviewer, the "Conclusion" section has been restructured (page 10, lines 436-445).
17) line 393-394: "break or at least maintain the mobility" what does it mean to break the mobility?
Reply
Dear Reviewer, this sentence has been deleted because the "Conclusion" section has been rewritten.
18) line 401-402: "On the other hand, it is worth noting that health promotion also depends on creating specific spaces for older adults to practise physical exercises and investments in the training of professionals to supervise physical exercises." Why is this mentioned? This study has not investigated health promotion via specific spaces or educating professionals
Reply
Dear Reviewer, we agree with your observation, the sentence was deleted.

Reviewer 3 Report
The research problem is relevant, but it needs some improvements, especially regarding the clarity of the methods and results. I have some comments below.
1. I think that exists a conceptual problem in your research. You have included in this analysis two variables that are conceptually similar. Which is the difference between “Physical Function” and “Physical Activity”?. This variable has been measured with different methods but shows similar characteristics. The physical function includes walk test (6-minute walk test/m), and you compare this to gait speed assessed by another walk test (50-foot Walk Test). It seems to me that you are evaluating the same concepts. If you can explain this better, your research will become clearer.
2. Your statistical analysis is not justified. Hypothesis and correlation tests seem to be unimportant in the mediation analysis phase. Why did you split the sample into two groups? Why did you do a correlation analysis?
3. You need to make your results clearer! It is very difficult to find a connection between what you are saying in the first paragraph of the discussion with what is described in the results. For example, “For PA-total, the analysis showed that a high GS was partially mediated by approximately 19% by better PF performance”. Where is this value? This is repeated several times in the same paragraph. You need to better explain your results within the results section. The model values you present must have meaning for the reader.
Minor points
Did you use the Version of modified Baecke Questionnaire for older adults? Did you use the Brazilian version?
Was all test performed at the same time? Consecutive physical tests can tire the elderly! If the answer is "yes” this is a bias, and you must identify it.
It is recommended to describe the analysis collection spaces and if there was any standardization about the clothes and shoes of the elderly. In this population, these two factors may interfere with the performance of the walk tests. If this has not been observed, identify it as a possible bias.
Author Response
Dear Reviewer, we are grateful for all the comments, and are available for future clarifications and/or corrections.
* Changes were made in the text using Microsoft Word's built-in track changes function.
1. I think that exists a conceptual problem in your research. You have included in this analysis two variables that are conceptually similar. Which is the difference between “Physical Function” and “Physical Activity”?. This variable has been measured with different methods but shows similar characteristics. The physical function includes walk test (6-minute walk test/m), and you compare this to gait speed assessed by another walk test (50-foot Walk Test). It seems to me that you are evaluating the same concepts. If you can explain this better, your research will become clearer.
Reply
Dear Reviewer,
1) We assume for the present study that Physical Activity and Functional Fitness are two different concepts, therefore:
a) We understand that Physical Activity is any bodily movement produced by skeletal muscles that requires energy expenditure. It considered the entire volitional movement of the body in space intentional behavior Caspersen et al. Public Health Reports, 100:2: 126-131, 1985;
b) Whereas the performance of Functional Fitness is having the physiological ability to perform normal daily activities safely and independently without undue fatigue (Rikli and Jones, 1999).
2) Physical activity was assessed by questionnaire. Functional fitness was evaluated by specific functional tests (SFT).
3) There is a difference between the tests: The 6-minute walk test (minutes/m) aimed to assess aerobic endurance. While the 30-foot (9m) walk test, performed at the participant's preferred and maximum speed, was used to assess their mobility. This information is now clearer, after rewriting the "2.2.5 Physical Function" section.
2. Your statistical analysis is not justified. Hypothesis and correlation tests seem to be unimportant in the mediation analysis phase. Why did you split the sample into two groups? Why did you do a correlation analysis?
Reply
Dear Reviewer, for a better understanding of the questions we review the Statistical analysis section (page 4, from line 213; and page 5, 221-225).
3. You need to make your results clearer! It is very difficult to find a connection between what you are saying in the first paragraph of the discussion with what is described in the results. For example, “For PA-total, the analysis showed that a high GS was partially mediated by approximately 19% by better PF performance”. Where is this value? This is repeated several times in the same paragraph. You need to better explain your results within the results section. The model values you present must have meaning for the reader.
Reply:
Dear Reviewer, thank you for your observation, the percentage values for each model were presented in the results section, as follows:
1) PA-total (page 7, lines 294-395);
2) PA-housework (page 7, lines 307-308);
3) PA-sport (page 8, lines 318-319);
4) PA-leisure (page 8, lines 330-31).
Minor points
1. Did you use the Version of modified Baecke Questionnaire for older adults? Did you use the Brazilian version?
Reply
Dear Reviewer, we used a Brazilian version of physical activity questionnaire for the elderly. Correct presentation of the questionnaire as well as references have been inserted (page 4, lines 171-173).
2. Was all test performed at the same time? Consecutive physical tests can tire the elderly! If the answer is "yes” this is a bias, and you must identify it.
Reply
Dear Reviewer, all questionnaires and functional fitness tests were carried out on the same day. All the assessments were scheduled in the morning, in small groups. For the most efficient use of the time and to minimize the fatigue effect for participants, testing stations were set up in a circuit-style in the following order, as defined by the protocol: (1) chair stand test, (2) arm curl test; (3) chair sit and reach test; (4) back scratch test, (5) 8-foot up-and-go test, and (6) 6-minute walk test.
3. It is recommended to describe the analysis collection spaces and if there was any standardization about the clothes and shoes of the elderly. In this population, these two factors may interfere with the performance of the walk tests. If this has not been observed, identify it as a possible bias.
Reply
Dear Reviewer, we tried to better describe the test we performed. However, for space reasons and to make the text shorter for the readers, we recommended an additional reading of the original reference for SFT manual, where a detailed description of the evaluation procedures, namely, equipment, procedures, scoring, and safety precautions.

Round 2
Reviewer 2 Report
To authors:
The authors made significant improvements to the manuscript, however I still have concerns regarding the manuscript. In the authors reply, there are numerous errors in the line referrals, which makes reviewing the manuscript more difficult. My main issue is that there are still erroneous citations, which I consider serious. Below is my specific feedback, written in bold font:
Minor:
Page 4, line 161: 2.2.3 Mental state
consider cognitive function or other term, as mental state could be interpreted as depressive symptoms etc.
Major:
On page 10, line 422-423, the authors write: “The cross-sectional design used in this study is a limitation since it does not allow the findings to be generalized to longitudinal changes in other populations. On the other hand, it should be considered that the results were accompanied by a formal analysis of mediation, considered a robust approach capable of inferring potential causalities.”
The first sentence states that the design (cross-sectional) does not allow generalizations to longitudinal changes, followed by a sentence that the mediation analysis however enables inferring potential causalities. Using “on the other hand” infers that the second sentence counters the argument of the preceding sentence, which I think it does not do. I assume the authors means that cross-sectional data can not support causal inferences, but that the mediation analysis enables causal inferences. The language cannot have these kinds of ambiguities.
The discussion is improved but still lacking. There are mentions of strengths and weaknesses in the first and last paragraph, and there are also issues of erroneous citations, e.g. number 48. Furthermore, the conclusion includes numerous new citations, where three citations in a sentence that discusses this study´s findings (page 10, line 436-439).
Dear Reviewer, we are grateful for all the comments, and are available for future clarifications and/or corrections.
Changes were made in the text using Microsoft Word's built-in track changes function.
Introduction
line 60: The authors state that PF is related to ADL, where "related to" implies a statistical association but the reference [1] is a qualitative study.
Reply
Dear Reviewer, we have replaced the reference with two review studies and meta-analysis (page 2, lines 59-60).
- Great
line 68-69: "All this enhances the establishment of a circle of unfavorable factors for mobility". This is an example of how almost all sentences in this paper can be made much shorter: this sentence can be replaced with: "All this impairs mobility". This is in itself not precise enough in my opinion but can be fixed with "All this impairs mobility in older persons"
Reply
Dear Reviewer, we appreciate your suggestion, the sentence has been adjusted (page 2, lines 68-69).
- Great
line 91-92 "Indeed, while the temporal parameters of life increase..." What does this mean? I could not find why the authors cited ref #22 for this sentence.
Reply
Dear Reviewer, the sentences have been corrected, as well as the references updated with longitudinal studies (page 2, lines 91-93).
- Great
line 99: "However, the causes of decreased GS and changes in its quality standards are not entirely clear" What are "quality standards?"
Reply
Dear Reviewer, the expression "quality standards" has been replaced by "gait standard". Moreover, clear examples of "gait patterns" have been included (page 2, lines 98-100)
- Great, though you misspelled the name of the first author in ref #30
Method
1) line 123: "this research" unclear what this means? The SEVAAI research project or this particular study?
Reply
Dear Reviewer, the sentence has been corrected (page 3, line 123).
- Great
2) line 132: "For the present study, four participants were excluded due to Parkinson's disease (n=1)" One or four participants excluded due to PD?
Reply
Dear Reviewer, the sentence has been corrected (page 3, line 139-143).
- Great
3) line 135-140: The authors state the criteria for the SEVAAI project and that 701 were included. They then state exclusion due to PD and AD, and a final sample of 697. After that they state inclusion criteria regarding age, ADL-independence, ability to perform assessments, no medical contraindication for physical activity, and exclusion critera of < 15 on the MMSE. With those criteria i would suspect that some participants would be excluded, but the final sample is still 697 to my understanding. The final sample size must be made clear, I suggest describing the SEVAAI first, with criteria and study sample, then a new paragraph with criteria for this particular study, and with the final sample stated last.
Reply
Dear Reviewer, Section 2.1. Study design and participants has been reorganized (page 3, line 121-143)
- Great, its now easy to follow.
4) line 155-156: "Strength in the lower part" Is this refering to lower extremities, i.e. legs, lower back, or both? please specify.
Reply
Dear Reviewer, we corrected the sentence using the expression "lower body strength" which is referred to in the Senior Fitness Test manual. We also clarify that the test is only intended to assess the strength of the leg muscles (quadriceps, glutes), (page 4, line 177-178).
- Great
5) line 157: "strength in the upper body (bar thread)" What is a "bar thread", a kind of exercise equipment? The authors should state which movement is performed, not the equipment used as for example a barbell can be used in a wide variety of movements.
Reply
Dear Reviewer, we replace the bar thread for a 2.3kg dumbbell for women and a 3.6kg dumbbell for men. In section 2.2.5, Physical Function, we explain better the movement in each physical fitness function test (please see page 4, from line 180-184).
- Great
6) line 160: abbreviation "SFT" is not established previously.
Reply
Dear Reviewer, "STF" abbreviation for Senior Fitness Test has been introduced (page 4, line 176)
- Great
7) line 166-167: "Participants were required to walk a distance of 50 feet at the preferred speed." Should it be "at their preferred speed"?
Reply
Dear Reviewer, thanks for the observation, we corrected the test distance as well as its execution mode (page 4, line 207-208)
- Great
8) line 186-188: "A complete mediation would be observed if the inclusion of objectively measured PF (mediator variable) reduced to zero the association observed between independent variable (PA-total, PA-housework" What would be reduced to zero? the p-value, correlation coefficient or no overlapping confidence intervals?
Reply
Dear reviewer, for better understanding the sentences have been revised and rewritten (page 5, lines 230-234). There is also an explanation on page 255, followed by a reference: Preacher, K.J.; Rucker, D.D.; Hayes, A.F. Addressing Moderate Mediation Hypotheses: Theory, Methods and Prescriptions. Multivariate Behavior. Res. 2007, 42, 185-227.
- Great
Results
1) line 211: "The high PA group attested..." consider using another word, as attested can imply that the participants only declared better performance.
Reply
Dear Reviewer, the word "attested" was changed to "indicated" (page 6, line 257)
- Great
2) line 213: "Members of the group with and high PA also indicated superior performance..." what does "with and high" mean? Also, that the group indicated superior performance is unclear, did they have better performance or not? in the latter case, the authors should also state that the group with higher PA had superior performance compared to the low PA group.
Reply
Dear Reviewer, the sentence has been corrected (page 6, line 258-260).
- Great
3) line 214: Please be concise with wording as to not confuse the readers. Exam could mean a written exam. The word "tests" usually works well.
Reply
Dear Reviewer, as requested, the word exam has been replaced throughout the text with test (page 6, line 260).
- Great
4) Page 5, table 1:
Hypertension, vision and hearing impairment is only mentioned in this table, the method section must describe how these are diagnosed (cutoffs systolic/diastolic bp vision and hearing impairment assessment). The authors must also describe if these are used as adjustments in any analyses or only used as descriptive statistics.
Reply
Dear Reviewer,
Information on collecting data concerning vision and addiction problems as well as blood pressure has been included in the section 2.2. We used all the information to better describe this sample (i.e., gender, age, years of education, falls, medication, visual and hearing impairment, and blood pressure) based on a health questionnaire employed in the FallProof! Program (Rose 2010) “Data collection (page 3, from line 147);
- a) Regarding blood pressure, the question was: “Have you ever been diagnosed as having high blood pressure? In this study, we didn’t analyze de scores of diastolic and systolic blood pressure.
- b) The explanation to readers that vision, hearing and blood pressure problems were only considered in descriptive statistics has been included in the Statistical analysis section (page 4, line 213)
- Great, very clear now.
5) Page 5 table 1, the row for male, column "low PA": 47.3 %: these are neither column% or row%, I guess it should be the former, but at 43.7%. Also, usage of male and female is uncommon, please consider "men, women".
Reply
Dear Reviewer, thanks for the observation. The percentage and denominations (men, women) were corrected in Table 1.
- Great
6) Mediation analysis presentations: The authors present beta-values with 2, 3 and 4 decimals, please revise to use the same number of decimals.
Reply
Dear Reviewer, we have adjusted all "beta-values" results to two decimals.
- Great
7) line 250: "Non-standard parameters." please explain what this means.
Reply
Dear Reviewer, an explanation of the use of parameters has been included at the end of the Statistical analysis section (página 5, linhas 250-251). We use beta values (β), therefore we present standardized parameters compared to CI intervals.
- Great
8) line 256: a P-value can never be exactly zero, use the "<" sign if the p value is very low and you want to avoid too many decimals
Reply
Dear Reviewer, we have adjusted the p-value as requested (section 3.4. Mediation analysis)
- Great
9) line 258-259: "a significant negative association was found between high PA-housework and low GS performance,β = 0.038", negative associations have a negative b-value, but this one have not.
Reply
Dear Reviewer, we corrected the result by adding the negative symbol. We also rounded the value from 0.038 to 0.04 (section 3.4, PA-housework)
- Great
10) line 278: "and the highest GS, β = .036;" is this the beta value of 0.03 in fig 5? a rounding error?
Reply
Dear Reviewer, the issue regarding the beta-value in text and Figure 5 has been fixed.
- Great
11) line 280: beta value of 0.017, is this the 0.01 in fig 5? a rounding error?
Reply
Dear Reviewer, the issue regarding the beta-value in text and Figure 5 has been fixed
- Great
Discussion:
1) The authors does not have a result summary, the structure is not good, different terms are used interchangeably (e.g., GS and gait, SB and physical inactivity)
Reply
Dear Reviewer, the first paragraph of the Discussion section has been restructured. Furthermore, the terms gait and physical inactivity were adjusted throughout the text for GS and SB, respectively.
- Great.
2) They state that e.g. "PF could fully mediate... by approximately 9%). It seems contradictory that something can fully mediate but still at only 9%. The authors should explain in more clear terms what these results mean.
Reply
Dear Reviewer, we agree with your suggestion. Therefore, we have corrected the term fully to partially throughout the text.
- Great.
3) line 293: "The premise" Do the author mean hypothesis? In that case, no such hypothesis was stated previously.
Reply
Dear Reviewer, the sentence has been deleted, and we have also restructured the first paragraph of the Discussion section (page 9, 335-350)
- Great.
4) line 294-296: high GS was partially mediated (19%) by better PF performance, and when PF was placed as a mediator, no effect was present. I dont understand, wasn't PF the mediator in the first sentence, but then the authors states "when PF was placed as a mediator"? This must be made clearer.
Reply
Dear Reviewer, we have corrected the information in the first paragraph (page 9, line 338-340).
- Great.
5) line 311, ref 44: this is a reference to the 2010 guidelines, but the WHO released updated guidelines in 2020 that should be cited instead.
Reply
Dear Reviewer, we have updated the reference [48].
- I could not find in the cited reference [48] that support your claim regarding only 25% of older adults reaching recommended levels of MVPA.
6) line 322: "studies" implies that the authors cited several studies, but I see only one.
Reply
Dear Reviewer, we have corrected the sentence, which should remain singular (page 9, line 371).
- Great.
7) line 328: "physical inactivity measured by PA housework", are the authors saying that housework is inactivity? If so, there are plenty of studies investigating MET values for common housework that the authors should read and revise this sentence.
Reply
Dear Reviewer, we corrected the sentence (page 9, lines 376-378).
- Great.
8) line 329-330: This is just describing the Beacke questionnaire and therefore should be placed in the method section.
Reply
Dear Reviewer, we have corrected the sentence, the detailed information from the Baecke Questionnaire has been removed, and adjusted in the Physical activity section (page 4, from line 167).
- Great.
9) line 331-332: "how many people are benefited" what does this mean?
Reply
Dear Reviewer, as clarified, this passage has been removed from the text.
- Great.
10) line 333: "All this reflects the mobility." Mobility of what, or who?
Reply
Dear Reviewer, as clarified, this passage has been removed from the text.
- Great.
11) line 335-336: The authors state that high levels of PA and GS are essential to perform ADL safely, but they cite two cross-sectional studies. Crossectional studies are unable to provide evidence of causality. I do not question the premise that PA and GS are associated, and likely also in a causal way, but the authors should be able to find RCTs or at least longitudinal studies that support this claim.
Reply
Dear Reviewer, as requested, we have included two systematic review studies and one longitudinal study (page 9, lines 380-381)
- Great.
12) line 338: "to present qualitatively mobility capacity" unclear what this means.
Reply
Dear Reviewer, the phrase has been corrected, using the expression more stable gait pattern (page 9, lines 383)
- Great.
13) line 338-339: "Gait is also a strong indicator of health and quality of life [9]. In this context, a poor GS is highly associated with an increased risk of falls [48,49,50]." Why do the authors use "In this context..." They mention health and QoL in the sentence before, which doesnt really translate to risk of falls.
Reply
Dear Reviewer, the meaning of the sentence has been corrected (page 9, lines 383-386)
- Great.
14) line 359: "southeast region" does that mean southeast region of Brazil? Then it can be shortened to "in south-east Brazil".
Reply
Dear Reviewer, thank you, the suggestion was followed (page 10, line 406)
15) line 371: "adduct", should it be adult?
Reply
Dear Reviewer, the sentence has been corrected (page 10, line 418).
- Great.
16) line 389-391: "The combination of endogenous factors related to vulnerable physiological aging [50,61] with low PA levels contributes to a low PF level, potentiating a vicious circle capable of progressively or abruptly reducing GS." This sentence is long and hard to follow, and a conclusion usually doesnt involve any citations as it should summarize the conclusions of the present study, not other studies.
Reply
Dear Reviewer, the "Conclusion" section has been restructured (page 10, lines 436-445).
- Great. However, the conclusion still contains citations. Furthermore, why does the second sentence state “Our findings highlight…” but with citations to other studies, when the author clearly talks about their own results?
17) line 393-394: "break or at least maintain the mobility" what does it mean to break the mobility?
Reply
Dear Reviewer, this sentence has been deleted because the "Conclusion" section has been rewritten.
- Great.
18) line 401-402: "On the other hand, it is worth noting that health promotion also depends on creating specific spaces for older adults to practise physical exercises and investments in the training of professionals to supervise physical exercises." Why is this mentioned? This study has not investigated health promotion via specific spaces or educating professionals
Reply
Dear Reviewer, we agree with your observation, the sentence was deleted.
- Great.
Reviewer 3 Report
The suggestions were sufficiently answered. Therefore, I recommend publishing the research. Congratulations.